# Do Children with Attention-Deficit/Hyperactivity Disorder Follow a Different Dietary Pattern than That of Their Control Peers?

**DOI:** 10.3390/nu14061131

**Published:** 2022-03-08

**Authors:** Meritxell Rojo-Marticella, Victoria Arija, José Ángel Alda, Paula Morales-Hidalgo, Patricia Esteban-Figuerola, Josefa Canals

**Affiliations:** 1Nutrition and Mental Health Research Group (NUTRISAM), Rovira i Virgili University (URV), 43201 Reus, Spain; meritxell.rojo@urv.cat (M.R.-M.); victoria.arija@urv.cat (V.A.); paula.morales@urv.cat (P.M.-H.); patricia.esteban@urv.cat (P.E.-F.); 2Research Centre for Behavioral Assessment (CRAMC), Department of Psychology, Rovira i Virgili University (URV), 43007 Tarragona, Spain; 3Institut d’Investigació Sanitaria Pere Virgili (IISPV), Rovira i Virgili University (URV), 43204 Reus, Spain; 4Attention Deficit Hyperactivity Disorder Unit, Sant Joan de Déu Hospital, 08950 Barcelona, Spain; joseangel.alda@sjd.es; 5Children and Adolescent Mental Health Research Group, Institut de Recerca Sant Joan de Déu, 08950 Barcelona, Spain

**Keywords:** attention deficit hyperactivity disorder, dietary patterns, school-age children, preschoolers, ADHD, food consumption

## Abstract

Attention-deficit/hyperactivity disorder (ADHD) is one of the most common neurodevelopmental disorders in children and adolescents. A current area of interest is the association between ADHD and food consumption. The aim of this study was to determine the food consumption and dietary patterns of children with and without ADHD in relation to their age and ADHD presentation. The study involved 259 preschoolers aged 3 to 6 years old (57 with ADHD and 202 controls) and 475 elementary-school-age children, aged 10 to 12 years old (213 with ADHD and 262 controls) from Spain. ADHD was diagnosed in accordance with the Diagnostic and Statistical Manual of Mental Disorders (5th edition) from Schedule for Affective Disorders and Schizophrenia for School-Age Children interviews. Eating data were collected using a food consumption frequency questionnaire, and principal component analysis was carried out to analyze dietary patterns. Western-like, sweet, and healthy patterns were identified. The ADHD group was negatively associated with the healthy pattern (*p* < 0.001) and positively associated with the Western-like diet (*p* = 0.004). Children with inattentive presentation showed lower adherence (12.2%) to a healthy pattern than that of the control group (39.9%) (*p* < 0.001). There is an association between ADHD and dietary habits; children with inattentive presentation may particularly be at risk of unhealthy eating habits.

## 1. Introduction

Attention-deficit/hyperactivity disorder (ADHD) is one of the most common neurodevelopmental disorders in children and adolescents. A recent systematic review established the global prevalence of ADHD to be between 2% and 7% [1]. However, depending on the study, this prevalence changes due to differences in age [2], sex [3], assessment methodology [4], and geographical location [3]. In Catalonia (Spain), one study that used administrative data for children aged 4 to 17 years estimated a prevalence of 4.06% [3], whereas in another study that looked at the school population, the rate was estimated to be 5.5% (3% for preschool children and 7.7% for school-age children) [2]. The prevalence of ADHD is higher in boys than that in girls, with a sex ratio of 2:1 in most studies [1,2,3].

The cause of ADHD is unclear. While genetic factors [5] play an important role, several environmental factors in the early stages of life can also alter neurodevelopment and increase the risk of ADHD [6,7]. Starting in pregnancy, nutrition is an important environmental risk factor for ADHD. Association was found between ADHD, and the mother’s altered nutritional and metabolic status [8]. In contrast, a healthy diet without ultraprocessed products [9] and a diet with a low inflammatory index (DII) during pregnancy [10] are associated with a lower incidence of ADHD in offspring. 

Associations between nutritional status and ADHD in children have also been analyzed. In this regard, dietary factors have been examined for their possible role in the etiology and treatment of this disorder. Some studies found altered levels of certain oligo elements such as magnesium, iron, and zinc in serum in children with ADHD [11,12,13,14,15]. For these reasons, several studies attempted to find a more natural, as opposed to pharmacological, treatment for ADHD; hence, various systematic reviews analyzed the effect of different nutritional interventions on behavioral symptomatology [13,16,17,18,19,20,21,22,23], but were unable to find clear supporting evidence. Furthermore, other authors studied the relationship between diet and ADHD versus non-ADHD. These studies used dietary patterns to evaluate the nutrition of children and adolescents with ADHD, and found relationships between ADHD and unhealthy diets [24,25], such as the fast food, sweet, and Western [26,27,28,29,30] patterns. This means that children and adolescents with ADHD prefer superfluous food with greater palatability and lower nutritional quality. They have a higher intake of unhealthy high-sugar and high-fat food, and a lower intake of fruits, vegetables, wholegrain cereals, and quality protein foods. This diet may contribute to the obesity risk found in samples of children with ADHD. A relationship was also found between obesity and ADHD, with obesity in children regarded as when the body mass index (BMI) is above the 95th percentile for age and sex [31,32], and in adults when the BMI is equal or greater than 30 kg/m^2^ [33]. In Spain, two studies specifically explored the association between ADHD diagnosis and the Mediterranean diet, a healthy dietary pattern that includes great diversity and high frequency of fiber-rich food such as fruits and vegetables, healthy fatty acids, proteins, and cereals of high nutritional quality [34,35]. They found lower adherence to the Mediterranean diet in children and adolescents with ADHD than that in controls, and less healthy habits in the eating process. However, these and other studies either do not provide data in terms of age periods or subtypes of ADHD presentation (inattentive, hyperactive–impulsive (H–I), or combined), or the sample size was not large enough. In addition, they did not consider factors that may influence the diet, such as socioeconomic level [36], psychological comorbid (internalizing) problems with ADHD [37,38], sex differences [6], pharmacological treatment [39], intelligence quotient [40], and ASD comorbidity [38].

In the present study, we aim to determine the type of diet in a sample of Spanish children with ADHD from two age groups: preschool and elementary school age. First, we analyze the food consumption of children with ADHD in comparison with their control peers and with the dietary guidelines for their age group. Second, we describe the dietary patterns of children with ADHD in relation to ADHD presentations in comparison with those of the control participants. We study differences according to age and the presentation of ADHD in relation to variables that may affect their eating habits, such as sex, parents’ educational level and profession (PELP), pharmacological treatment, intellectual quotient (IQ), internalizing problems, and the presence of any autism spectrum disorder (ASD) comorbidity. We hypothesize that children with ADHD have a low-quality diet with a lower intake of fruits and vegetables than that of their peers without the disorder, and that this shows up specifically in children with combined or hyperactive–impulsive (H–I) ADHD presentations. Moreover, we hypothesize that food consumption generally differs from the established recommendations by dietary guidelines.

## 2. Materials and Methods

### 2.1. Study Design and Participants

The participants in this study come from two research projects carried out in Spain: the Neurodevelopmental Disorders Epidemiological Research Project (EPINED) and the characterization of the metabolic profile (bacterial and nonbacterial) in children and adolescents with ADHD and evaluation of its value as a diagnostic marker (MetigenADHD). EPINED was a two-phase study carried out between 2014 and 2019 with the main aim of estimating the prevalence of ADHD and autism spectrum disorder (ASD) in the school population (preschoolers and elementary-school children) of the province of Tarragona, Catalonia, Spain. In the first phase, the risk of both disorders was screened by validated tests administered by parents and teachers (*n* = 3727). In the second phase, children at risk and a control group without risk (781 children; 259 preschoolers) were individually assessed by trained clinicians to confirm ADHD and ASD diagnoses according to the criteria of the Diagnostic and Statistical Manual of Mental Disorders (5th edition) (DSM-5). Participants received a positive diagnosis of ADHD (i.e., clinical ADHD) when information in the K-SADS-PL interview met the DSM-5 criteria for any of the three presentations of ADHD (inattentive, hyperactive–impulsive, and combined). Two children who had been diagnosed were considered to have clinical ADHD, although they were in partial remission from symptoms due to pharmacological or psychological treatment. Participants were considered to have subclinical ADHD when they scored positive on parent and teacher Conners’ 10-item indices (T ≥ 65) and presented on the Schedule for Affective Disorders and Schizophrenia for School-Age Children (K-SADS-PL) [41] four or five manifestations for any of the ADHD presentations (inattentive or hyperactive–impulsive) with considerable impact on their functioning. Further descriptions of the EPINED project can be found in Canals et al. [2], and Morales, Voltas, and Canals [42]. The MetigenADHD study was carried out between 2017 and 2019 in Barcelona (Catalonia, Spain) with the aim of evaluating the role of metabolites present in feces and urine as diagnostic markers of ADHD in children and adolescents. In this study, children with an ADHD diagnosis were recruited by the ADHD unit of Sant Joan de Déu Hospital (Barcelona). The study was offered to families who were accessing the unit for the first time and who were diagnosed de novo or confirmed in relation to previous data. The total sample of the current report comprised 210 children with ADHD (42 preschoolers and 168 from elementary school), 60 with subclinical ADHD (15 preschoolers and 45 from elementary school), and 464 controls, who were children without an ASD or ADHD diagnosis (202 preschoolers and 242 from elementary school). 

### 2.2. Psychological Assessment

For the ADHD diagnosis, the K-SADS-PL semistructured diagnostic interview was administered to the parents. Two instruments were used to diagnose ASD: the Autism Diagnostic Observation Schedule (2nd edition) (ADOS-2) [43] and the Autism Diagnostic Interview Revised (ADI-R) [44]. The parents of all the children answered the Child Behavior Checklist (CBCL) [45] to obtain information on psychological and emotional problems (externalizing, internalizing, and total problems). To estimate overall intelligence (IQ), the Spanish versions of the Wechsler Scales of Intelligence for preschool (WPPSI-IV) and elementary-school (WISC-IV) children were administered. All the administered tests had good psychometric reliability and validity properties [46,47,48]. The tests were administered in person, and parents and children answered separately. All diagnoses were by trained psychologists and psychiatrists. Further information about the psychological assessment methodology of the EPINED project can be found in studies by Canals et al. [2], and Morales, Voltas, and Canals [42].

### 2.3. Nutritional Assessment

Data on the participants’ eating habits were collected by their parents using two different food consumption frequency questionnaires (FCFQ), with one validated for preschool children (41 items [49]) and another validated for elementary-school children (45 items [50]). These questionnaires gave us information about servings per week and per month. The grams per day of each item were calculated in relation to the age of the participants and the size of the recommended ration as stipulated by the regional health authority [51] and in agreement with the experts in this field in our research group [52,53,54]. 

The dietary guidelines of the Spanish Society of Community Nutrition (SENC) [55] were used to compare the frequency of food consumption (servings per day or per week) with the recommended frequency of consumption for children. The studied food groups (and subgroups) are presented in Appendix A.

The Spanish Quality Diet Index (SQDI) [56] was used. This index includes nine food groups on the basis of their nutritional quality. Therefore, meat, fish, and eggs were regrouped into a single group, and sweets and sweet cereals were also grouped into a single group. The index compared recommended servings with real consumption and gave a score of between 0 and 100 points. Lastly, these scores were classified into three categories: ≥80, healthy; 50–79, needs to improve; and ≤49, unhealthy [56].

Principal component analysis (PCA) following general procedures was used to identify the dietary pattern [57]. 

### 2.4. Other Variables

#### 2.4.1. Anthropometric Measurements

A dietitian certified by the International Society for the Advancement of Kinanthropometry (ISAK) took the anthropometric measurements. A SECA^®^ stadiometer, accurate to 0.1 mm (PERILB-STND), was used to measure height (cm), and TANITA scales (BC 420SMA) were used to measure weight (kg). Then, body mass index (BMI)(Kg/m^2^) and the BMI z score were calculated on the basis of the World Health Organization Child Growth Standards [58]. 

#### 2.4.2. Sociodemographic Data and Parents’ Educational Level and Profession

Sociodemographic data were reported by parents, and parents’ education level and profession (PELP) were calculated by adapting the Hollingshead [59] index formula.

### 2.5. Statistical Analysis

To describe the study sample, variables were compared by age group (preschool and elementary school) and diagnosis (clinical ADHD, subclinical ADHD, and controls). 

ANOVA was used to analyze food intake in grams for each diagnosis (clinical, subclinical, and control) and for each age group. No differences were found between diagnosis (clinical and subclinical) or presentations, so the ADHD group was created from clinical and subclinical subgroups. For the food intake per servings analysis, in addition to the diagnosis groups, we combined the two age groups. In this case, the number of servings is the same for both age groups but the grams per intake is not. After combining the groups for both analyses, Student’s *t*-test was used instead of ANOVA. Lastly, we carried out ANCOVA, adjusting for our control variables. 

PCA was used to identify the dietary patterns. First, FCFQ items were grouped into the 20 food groups presented later in the Results section. In the analytical process, we used similar parameters to those in other studies of dietary patterns [57]. Consumed grams were standardized in each food group, so that they all had the same weight in the analytical item given their different average consumption values. Analysis gave an eigenvalue for each component. Those with an eigenvalue >1 were factors to be extracted. A Scree plot test was generated to confirm the number of factors to retrain. A factor loading matrix was also generated to extract weights (factor loading) for each analyzed food group. Food groups with factorial load ≥ 0.30 were regarded to be important contributors to the dietary patterns. Dietary pattern score was generated for each pattern and participant; these variables were calculated as linear combinations of the standardized intake of the 20 food groups weighted by their factor score coefficients, which were automatically generated by the statistical software. With this method, all adolescents received a score for the 3 measured dietary patterns. Then, the dietary pattern scores were categorized into tertiles. ANOVA and Student’s *t*-test analysis were performed to assess associations between the dietary patterns and the different groups in diagnostic categories (a) ADHD diagnoses (ADHD vs control) and (b) with ADHD presentation (inattentive, hyperactive–impulsive (H–I) and combined). ANCOVA was used to adjust for covariates that may be related to food consumption: sex, PELP, internalizing problems, pharmacological treatment, IQ, and ASD comorbidity. Chi-squared analyses were performed to identify the level of participant adherence to each dietary pattern. Control for multiple testing with Bonferroni correction was performed within the ANOVA, ANCOVA, and chi-squared tests. A 95% confidence interval was provided for all estimations. Statistical analyses were performed using IBM SPSS 27.

## 3. Results

### 3.1. Study Sample Description

Table 1 presents the sociodemographic, psychological, nutritional, and anthropometrical characteristics per age group and ADHD diagnosis. Preschoolers were aged from 3 to 6 years old, and elementary schoolers were aged from 10 to 12 years old. Participants were 210 with ADHD, 60 with subclinical ADHD, and 464 controls. Most of the sample was Spanish and had a medium score for parents’ educational level and profession. No differences were found in the BMI and BMI z scores in either of the two age groups. Although the score was within the mean range, we found significantly lower IQ in the preschoolers and elementary group with ADHD in relation to the control group. A total of 34.5% of the elementary school-age children with ADHD were receiving pharmacological treatments, and 100% of these were stimulant drugs. Internalizing problems measured by the CBCL showed higher scores in both the clinical and subclinical groups compared to the control group for both age groups.

### 3.2. Food Consumption

Food consumption in grams and servings was calculated for the three diagnosis groups, and no differences were shown between the clinical and subclinical groups, or between presentations of ADHD. For these reasons, food consumption was recalculated after combining the ADHD groups (ADHD and subclinical ADHD, and their presentations). Appendix A shows these results in grams per day, analyzed for each age group. In preschoolers, only preserved fruit showed clearly higher consumption in the ADHD group. When analysis was adjusted with covariates (ANCOVA), results did not change (see Appendix A).

To compare food consumption with SENC recommendations, the two age groups were combined because the servings were the same for both. Figure 1 and Figure 2 show the food consumption in servings per day and per week, respectively. Significant differences were found in fruit consumption (*p* = 0.033), which was higher in the control group. In both graphs, minimal and maximal servings recommended by the public administration are expressed by dots and asterisks, respectively, except for those foods that should not be eaten or should be eaten occasionally, for which there is no recommendation. Only the consumption of protein foods in general, dairy products (milk, yogurt, and cheese), and fish agreed with the experts’ recommendations.

After calculation, the SDQI did not show any differences in either of the two age groups. In preschool children, 95.20% of the children with ADHD and 96% of the control group needed to improve their diet quality (*p* = 0.905). In the elementary-school-age group, 92.30% of the children with ADHD and 92.40% of the control group also needed to improve (*p* = 0.992) (for more results, see Appendix A).

### 3.3. Dietary Patterns

Principal component analysis was performed for each age group. Three dietary patterns were identified with the same consumption trend for the two age groups. In this way, and to obtain more consistent results, analysis was repeated by combining the two age groups. Table 2 shows the factor-loading matrix with the three main dietary patterns created by principal factor analysis with a total variance of 32.103%. Food groups where the loading score was ≥0.3 in more than one dietary pattern were assigned to the pattern with the highest loading score.

The first pattern was tagged as Western-like, and was formed by eggs, white, red, and processed meat, seafood, savory cereals, potatoes, legumes, sodas, and cooked vegetables. The second pattern was tagged as Sweet, formed by dairy desserts, sweet cereals, preserved fruit, and sweets. The last one was tagged as Healthy and included nuts, fish (white and oily), raw vegetables, fresh fruit, and olive oil. Lastly, they were ranked and converted into tertiles to analyze adherence.

Table 3 shows dietary pattern scores in mean (SD) for the diagnosis and ADHD presentations. After adjusting for our control variables, the Western-like pattern showed a higher score in the ADHD group than that of the control group (*p* = 0.004). Among presentations, only ‘inattentive’ had a higher score than that of the control group (*p* = 0.004). The Healthy pattern had a higher score in the control group than that in the ADHD group (*p* < 0.001). Table 4 shows the adherence to these patterns. The control group had a higher adherence (39.9%) to the Healthy dietary pattern (*p* < 0.001) than the ADHD group did (22.2%). Among the presentations, the inattentive type showed the lowest adherence (45.9%) to that pattern (*p* < 0.001). In contrast, no significant differences were found for the Western-like and sweet dietary patterns.

## 4. Discussion

This study provided us with knowledge about how children with ADHD and its presentations in two age groups eat in comparison with their control peers and about their adherence to the dietary guidelines. It allowed for us to demonstrate that children with ADHD, and especially the inattentive presentation, had a less healthy diet than the children in the control group did. However, all participants needed to improve their diet because they did not follow the established dietary recommendations.

We considered two age groups on the basis that they would present differences in the consolidation of habits. Furthermore, in young children such as preschoolers, there is usually a higher proportion of hyperactivity, which decreases with age [2], while inattention persists throughout life. We thought that these characteristics could affect eating behavior depending on age. However, after analysis, we did not find any remarkable differences between age groups either in terms of food intake (by grams or servings) or between children with ADHD and their control peers. Results did not change after adjusting for our control variables (sex, PELP, internalizing problems, pharmacological treatment, IQ, and ASD comorbidity). In contrast, Rios-Hernandez et al. [34], in their study on the Mediterranean diet and ADHD, found that children with ADHD had lower consumption of vegetables and oily fish, and a higher consumption of sugar, candy, and cola and noncola soft drinks than children in the control group did. This difference in sweet consumption could be due to the fact that our sample was much larger, and included both clinical cases and school children. It provided more diversity in relation to symptom severity which may affect our results by attenuating the observations regarding sweet consumption. We also thought that another reason was the lack of medicated subjects in the Rios-Hernandez et al. study. Children with ADHD have a chemical imbalance of dopamine and noradrenaline neurotransmitters, and a dysfunction in the regulatory circuits between the prefrontal cortex and the basal ganglia, which actively participate in executive functions and are related to the reward system [60]. Stimulant treatment increases this dopamine by blocking its transporters, resulting in an increase in this neurotransmitter. However, although a total of 36% of the elementary group in our study were medicated with stimulant drugs, after applying the control variables (including pharmacological treatment), sweets consumption did not change. 

Regarding the daily and weekly servings of food consumption, our sample only respected the SENC [55] recommendations for three items: protein products, dairy products, and oily fish. Those intakes were adequate in both the ADHD and control groups. Consumption of the rest of the items in both groups were far above the recommended amounts (such as fruit juice, dairy desserts, meats, especially red meat, sweet cereals, and sweet sodas) or far below (such as starch, vegetables, fruits, legumes, nuts, white meat, white fish, and eggs). This is linked to the low scores observed in the SDQI for the entire sample, and indicates the need to improve children’s diets in general. These results paint a picture of a current problem in many developed countries: nutritional deficiencies and excesses that can lead to childhood obesity. In 2019, a study on nutrition, physical activity, child development, and obesity (Aladino) in Spain showed that 23.3% of children between 6 and 9 years were overweight, and 17.3% were obese [61]. In Europe, the European Childhood Obesity Surveillance Initiative (COSI) carried out in 2017 by the World Health Organization (WHO) showed that the Mediterranean countries (Cyprus, Greece, Italy, and Spain) had the highest rates of childhood overweight and obesity [62]. Although we were not looking for causality, we thought it was important to know if our participants were following dietary guidelines, because it offers information about their eating habits. If children with ADHD were the only ones who did not follow dietary guidelines, this would also give us valuable information.

Results of the food consumption comparisons did not provide any conclusions about the quality of our participants’ diets. The quality of the diet is not only determined by the consumption of some food or group of foods, but by way of eating while following a dietary pattern. Tucker [63] also stated that the establishment of dietary patterns is better and more useful for health promotion. In the study by Rios-Hernandez et al. [34], the Mediterranean diet was considered to be a dietary pattern itself, and was extracted directly from the KIDMED test. In contrast, our dietary patterns were established through factor analysis. Our methodology was used in several settings and is suitable for describing the usual dietary intake [64]. 

As far as dietary patterns are concerned, PCA results showed three main ways of eating, which we named Sweet, Healthy, and Western-like. In the latter case, it is important to say Western-like and not simply Western because this pattern also contains healthy foods, such as cooked vegetables and legumes. Traditionally, in this region, cooked vegetables are usually eaten with boiled potatoes and legumes with some processed meat, a fact that may drag this healthy food into the Western-like pattern. In line with studies by Abbasi et al. [27], and Howard et al. [28], the systematic review by Del-Ponte et al. [24], and the meta-analysis by Shareghfarid et al. [25], we found positive association between the Western pattern and ADHD. Yan et al. [29] found that unhealthy patterns such as processed food, snacks, and beverages were positively associated with ADHD symptomatology. Moreover, that this association was especially strong among those who presented inattention, much more than that among the other ADHD subtypes or the control group. In terms of adherence, although only a statistical trend was observed, the ADHD group had the highest adherence to the Western-like pattern. Furthermore, Henriksson et al., using data from the Healthy Lifestyle in Europe by Nutrition in Adolescence (HELENA) study, suggested that attention capacity could be influenced by healthy dietary patterns in adolescence, but that the study design meant that conclusions could not be drawn about the causality and the direction of the associations [65]. Furthermore, these consumption trends of the Western-like pattern also agree with Lee et al. [66], who found that meat and carbohydrate patterns were associated more with the inattentive kind of risk in children at 6 years of age. This association that we found for inattention in our study contradicts the previously stated hypothesis that there should be more possible differences between the ADHD and the control group in the H–I and combined presentations due to their clinical characteristics of less inhibitory control and greater self-regulation problems. However, our data support those found in adults, which demonstrated that inattentive symptoms of ADHD were also directly related to bingeing or disinhibited eating behavior, and indirectly to internal appetite signals by pathways of association via negative mood [67,68]. This emotional dysregulation may lead to emotional eating, which in turn may significantly influence the individual’s food choices (usually for unhealthy and comfort food). This food activates dopamine by acting as a stimulus for the reward system [37,60,69,70].

As far as the Healthy pattern is concerned, our results agree with those of Abbasi et al. [27], Dong Woo et al. [71], Yan et al. [29], Shareghfarid et al. [25], and Zhou et al. [72], who found a negative association between that pattern and ADHD. We also found that the Healthy pattern was positively associated with the control group, which adhered to this pattern much more than any of the ADHD presentations. Although both presentations showed significantly low adherence to this pattern, the inattentive one had the lowest preference for this way of eating. 

After analysis, we did not find any differences for the Sweet pattern between the ADHD and the control groups, or for the H–I and combined presentations, as we expected. As previously mentioned, the use of stimulant drugs could relieve the craving for sweets. In this regard, 72.88% of the medicated children (all with stimulant drugs) presented the H–I and combined presentation. However, we did not find any differences after adjusting for pharmacological treatment. In agreement with our results, Bowling et al. [30] found that their participants continued eating in an unhealthy way despite their medication. 

In contrast to our findings, Azadbakht and Esmaillzadeh [26] concluded from a cross-sectional study in Iran that the Sweet dietary pattern was associated with a higher prevalence of ADHD but did not find a positive or negative association between the Western or Healthy patterns and ADHD as we did. This disagreement could be due to food differences between cultures. Moreover, their study did not include preschool children, who do not have the same choice capacity as that of older children, who can buy food at the school canteen. In addition, our study sample was larger and had a higher percentage of medicated children.

### Strengths and Limitations

A strength of this study is the large number of participants composing a representative sample of the Spanish child population and the different ADHD presentations. Furthermore, psychological and nutritional assessment was performed by rigorous and validated methodologies. In terms of limitations, the fact that is not a longitudinal study did not allow for us to determine the directionality of the relation and see whether the diet helps or aggravates the symptomatology of ADHD. Although FCFQ is a great tool that gave us many useful data, a three-day record questionnaire and nutritional biochemistry profiles could give us additional and more realistic information about the children’s feeding process. In addition, this information was reported by external observers, these being either the parents or legal guardians because of the participants’ age. Elementary-school-age children, mostly those with ADHD, could consume unhealthy foods such as sweets and not tell their parents, meaning that it would not have been recorded, leading to a possible reporting bias.

## 5. Conclusions

The ADHD group, especially the inattentive presentation, adhered less to the Healthy pattern than the control group did, and it was positively associated with the Western-like pattern. The control group had the highest adherence to the Healthy pattern. However, the two groups had a similar adherence to the Sweet dietary pattern. This, the recommended servings comparison, and the SDQI results show that the study population generally has a poor-quality diet that it needs to improve to prevent future health issues. 

## Figures and Tables

**Figure 1 nutrients-14-01131-f001:**
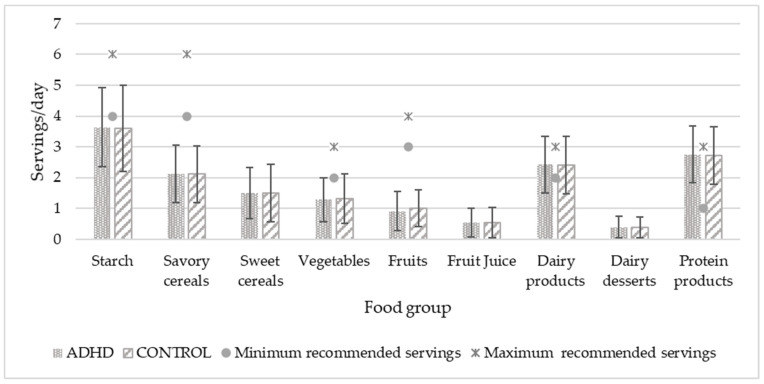
Daily food consumption of study population in relation to recommended servings. Servings/day values are expressed by mean and standard deviation. Significant differences in fruit consumption are higher in the control group with *p* = 0.033. *p* < 0.05.

**Figure 2 nutrients-14-01131-f002:**
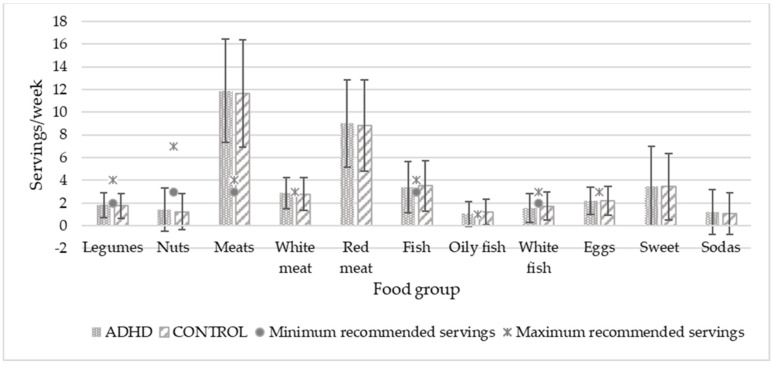
Weekly food consumption of study population in relation to recommended servings. Servings/day values are expressed by mean and standard deviation. No significant differences were shown. *p* < 0.05.

**Table 1 nutrients-14-01131-t001:** Sociodemographic, psychological, nutritional, and anthropometric characteristics per age group and ADHD diagnosis.

	Preschool-Age Children				Elementary School-Age Children			
	ADHD ^a^	Subclinical ^b^	Control ^c^	*p*	ADHD ^a^	Subclinical ^b^	Control ^c^	*p*
	*n* = 42	*n* = 15	*n* = 202		*n* = 168	*n* = 45	*n* = 262	
Age. Years *	5.18 (0.41)	4.90 (0.60)	5.09 (0.69)	0.364	11.02 (0.71)	10.98 (0.47)	11.11 (0.55)	0.208
Sex. Males	66.70 (28)	46.70 (7)	57.90 (117)	0.36	73.20 (123)	53.30 (24)	56.50 (148)	**0.001**
Origin (Spanish)	95.20 (40)	73.30 (11)	76.70 (155)	**0.002**	87.50 (147)	88.90 (40)	87.00 (228)	0.439
ADHD presentation								
Inattentive	7.10 (3)	-	-	**<0.001**	40.50 (68)	60.00 (27)	-	**<0.001**
Hyperactive–impulsive	31.00 (13)	53.30 (8)	-		7.70 (13)	13.30 (6)	-	
Combined	61.90 (26)	46.70 (7)	-		51.80 (87)	26.70 (12)	-	
PELP classification								
Low	23.80 (10)	40.00 (6)	15.30 (31)	0.099	19.00 (32)	22.20 (10)	14.90 (39)	0.677
Medium	64.30 (27)	46.70 (7)	64.40 (130)		62.50 (105)	57.80 (26)	65.30 (171)	
High	11.90 (5)	13.30 (2)	20.30 (41)		18.50 (31)	20.00 (9)	19.80 (52)	
BMI *	16.2 (2.36)	15.19 (1.20)	15.90 (2.16)	0.295	19.82 (4.00)	19.69 (3.88)	19.78 (3.89)	0.983
zBMI *	0.50 (1.39)	−0.12 (0.85)	0.30 (1.32)	0.294	0.75 (1.41)	0.74 (1.39)	0.77 (1.37)	0.993
IQ *	89 (15)	102 (16)	99 (15)	**<0.001**	96 (14)	99 (13)	104 (16)	**<0.001**
				**<0.001 ^ac^**				**<0.001 ^ac^**
				**0.018 ^ab^**				
ASD								
Subclinical	11.90 (5)	-	-	**<0.001**	6.50 (11)	2.20 (1)	-	**<0.001**
Clinical	14.30 (6)	-	-		7.10 (12)	-	-	
CBCL score *	65.93 (10.58)	64.33 (7.22)	56.27 (11.05)	**<0.001**	61.95 (9.41)	56.20 (8.37)	54.18 (9.87)	**<0.001**
				**<0.001 ^ac^**				**<0.001 ^ac^**
				**0.018 ^bc^**				**0.001 ^ab^**
Stimulant treatment	-	-	-		34.50 (58)	-	-	**<0.001**

ADHD, attention-deficit hyperactivity disorder; Control, children without ADHD; PELP, parents’ educational level and profession; BMI, body mass index; zBMI, BMI for age (z score); IQ, intelligence quotient; ASD, autism spectrum disorder. * Mean (SD); percentage (*n*) ANOVA was used for quantitative variables, and chi^2^ for qualitative. Significant differences in bold (*p* < 0.05). For subgroups, only significant *p* values are shown. Superscript letters of *p* values indicate in which subgroups significance was reduced.

**Table 2 nutrients-14-01131-t002:** Factor-loading matrix for three dietary patterns.

	Dietary Patterns		
	Western-Like	Sweet	Healthy
Dairy products	0.114	0.208	0.205
Eggs	**0.383**	0.020	0.330
White meat	**0.562**	−0.266	−0.138
Red and processed meat	**0.624**	−0.023	−0.215
Seafood	**0.408**	−0.080	−0.103
Savory cereals	**0.523**	−0.154	−0.267
Potatoes	**0.546**	0.112	0.089
Legumes	**0.533**	−0.323	0.009
Sodas	**0.435**	0.304	−0.286
Cooked vegetables	**0.476**	−0.457	0.260
Dairy desserts	0.257	**0.426**	−0.187
Sweet cereals	0.334	**0.606**	−0.007
Preserved fruit	0.260	**0.568**	0.009
Sweets	0.227	**0.537**	−0.042
Nuts	0.344	0.052	**0.390**
White fish	0.070	0.266	**0.489**
Raw vegetables	0.316	−0.194	**0.423**
Fresh fruit	0.268	−0.187	**0.388**
Olive oil	−0.293	0.216	**0.484**
Oily fish	−0.009	0.185	**0.277**
% of variance	14.903	9.753	7.446

Factor-loading highlighted in bold to show the food groups classification.

**Table 3 nutrients-14-01131-t003:** Dietary pattern by diagnosis and ADHD presentation.

	Total ADHD	ADHD Presentation		Control				
		Inattentive ^a^	H–I and Combined ^b^		*p* ADHD-Control	*p* ADHD-Control	*p* Presentations-Control	*p* Presentations-Control
	*n* = 270	*n* = 98	*n* = 172	*n* = 464	Raw	Adjusted ^1^	Raw	Adjusted ^1^
Western-like dietary pattern	222.89 (77.06)	234.52 (88.43)	216.26 (69.16)	209.46 (68.64)	**0.015**	**0.004**	**0.007**	**0.004**
							**0.005 ^ac^**	**0.003 ^ac^**
Sweet dietary pattern	68.91 (39.77)	67.12 (41.35)	69.93 (38.92)	68.54 (36.48)	0.897	0.537	0.834	0.697
Healthy dietary pattern	137.11 (59.56)	121.50 (55.78)	146.01 (59.96)	155.77 (64.05)	**<0.001**	**<0.001**	**<0.001**	**<0.001**
							**<0.001 ^ac^**	**<0.001 ^ac^**
							**0.006 ^ab^**	**0.001 ^ab^**

ADHD, attention deficit hyperactivity disorder; Control, children without ADHD; H–I, hyperactive–impulsive. Values in mean (SD). *p* ^1^ value adjusted for sex, PELP, pharmacological treatment, IQ, internalizing problems (CBCL), and ASD comorbidity. Significant differences in bold (*p* < 0.05) For subgroups, only significant *p* values are shown. Superscript letters of *p* values indicate in which subgroups the significance is reduced.

**Table 4 nutrients-14-01131-t004:** Dietary pattern adherence per diagnosis and ADHD presentation.

	Total ADHD	ADHD Presentation		Control	*p* ADHD-Control	*p* Presentations-Control
		Inattentive ^a^	H–I and Combined ^b^			
	*n* = 270	*n* = 98	*n* = 172	*n* = 464		
Western-like dietary pattern					0.052	0.088
Low adherence	31.1% (84)	25.5% (25)	34.3% (59)	34.5% (160)		
Medium adherence	30.0% (81)	32.7% (32)	28.5% (49)	35.3% (164)		
High adherence	38.9% (105)	41.8% (41)	37.2% (64)	30.2% (140)		
Sweet dietary pattern					0.401	0.515
Low adherence	36.3% (98)	40.8% (40)	33.7% (58)	31.5% (146)		
Medium adherence	32.2% (87)	29.6% (29)	33.7% (58)	34.1% (158)		
High adherence	31.5% (85)	29.6% (29)	32.6% (56)	34.5% (160)		
Healthy dietary pattern					**<0.001**	**<0.001**
Low adherence	38.5% (104)	45.9% (45)	34.3% (59)	30.2% (140)		**0.009 ^ab^**
Medium adherence	39.3% (106)	41.8% (41)	37.8% (65)	30.0% (139)		**<0.001 ^ac^**
High adherence	22.2% (60)	12.2% (12)	27.9% (48)	39.9% (185)		**0.018 ^bc^**

ADHD, attention deficit hyperactivity disorder; Control, children without ADHD; H–I, hyperactive–impulsive. Values in percentage (*n*). Significant differences in bold (*p* < 0.05.) For subgroups, only significant *p* values are shown. Superscript letters of *p* values indicate in which subgroups significance is reduced.

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
