# Peer review of "Do Children with Attention-Deficit/Hyperactivity Disorder Follow a Different Dietary Pattern than That of Their Control Peers?"

_nutrients, 2022, doi:10.3390/nu14061131_

Round 1

Reviewer 1 Report

Derived from both a population-based and a clinical sample of pre-school and school-aged children, this study sought to elucidate the cross-sectional associations between diagnosis of clinical or sub-clinical attention-deficit/hyperactivity-disorder (ADHD) on the one hand and eating behaviour, specifically food consumption and dietary patterns on the other hand. Controlling for a range of socioeconomic and psychological health variables and compared to participants without a diagnosis of ADHD, those with ADHD and particularly those with the inattention subtype were less likely to show a healthy eating pattern and more likely to report a Western-style diet. The authors additionally compared the consumption of specific foods and food groups in their sample to the age-specific Spanish dietary recommendations and found low levels of adherence to the recommendations. The authors thus add evidence to the existing literature on the associations between ADHD symptomatology in children and adolescents and their eating behaviour. Limited by its cross-sectional design, the study recommends future research including prospective study designs to further elucidate causal relationships between ADHD symptomatology and eating behaviour.
While the study raises interesting research aims and accumulates results derived from both a population-based and a clinical sample, some methodological aspects remain unclear and therefore potentially reduce statistical and clinical significance of the findings. English language editing should be considered.

Comments (including line numbers)

Abstract:

  • Consider the term “neurodevelopmental” rather than “neurodevelopment” (12)
  • Please specify “young population” (13). Consistently provide information on the age ranges of both subsamples (15-17).
  • Briefly state how “dietary patterns were analyzed”, yielding in identification of the three eating patterns (20,21).
  • The findings regarding adherence to the eating patterns seemed rather redundant, since they basically have the same implication as the findings regarding the differences in prevalence of eating patterns between ADHD symptomatology groups. If not, please state their additional merit for the study aims (23,24).
  • In the last sentence of the abstract, consider specifying the term “symptomatology” (25,26) and thereby the important finding that only specific subtypes of ADHD (i.e. the inattention subtype) may be at risk for unhealthy eating patterns.

Introduction:

  • Please consider English language editing and spell checking for the whole manuscript.
  • Please further elaborate the stated differences in ADHD prevalence across “age, sex, assessment methodology and geographical location” (34,35). As this is far from an understudied area, please provide high-quality data regarding all confounding variables derived from systematic reviews and meta-analyses of large epidemiological samples.
  • Be more specific when describing aetiology (including relevant life-style or environmental factors) of ADHD. For example, give examples for the environmental factors increasing the risk for ADHD other than eating behaviour. Thoroughly describe the existing evidence for associations between ADHD symptomatology and nutritional aspects in general and, specifically, healthy or unhealthy eating behaviour, with regard to the investigated age groups.
  • Consider revising the sentence “Furthermore, other authors have studied the relationship between diet and ADHD in comparison with the diet of the population without ADHD.” (48,49). Please revise the terms “sugar-high” and “fat-rich” and consider changing them to “high-sugar” and “high-fat” (55).
  • In the sentence “In addition, a relationship has also been found between obesity and ADHD in children [27,28] and adults [29].” (56,57) please be more specific and, at least, define obesity according to the body mass index as well as its connection to the concepts of your study.
  • Please specify “ADHD presentations” when they are first mentioned (64).
  • Please specify the age groups (68).
  • Brackets of the second level should be formatted as square brackets (70).
  • How does analysing and comparing the different levels of adherence to eating recommendations between the clinical groups add information compared to the different prevalence rates of eating patterns in the clinical groups (71,72)? Please elaborate.
  • Do you have any hypotheses on your numerous control variables? In any case, provide existing evidence from which you have (or have not) derived your hypotheses (72-75).
  • Consider numbering your study aims and using that structure throughout the whole manuscript.

Methods:

  • In general, incorporating two samples derived from the general as well as a clinical population might strengthen external validity of the results. However, within your results, you do not account for this important aspect of sample characterisation. Since the participants from the second sample (the MetigenADHD study) were recruited from an ADHD treatment unit, their eating patterns might be influenced by the ongoing or completed therapy. Please give information for treatment completion rates in the second sample and state in detail, how you have controlled for effects of an ongoing therapy or other possible sample bias.
  • Within the sample description, please consistently add information on the age of participants for all stages of the sampling procedure.
  • You state that “Children with previous ADHD diagnoses provided by public mental health centers […] were also considered as having “clinical ADHD.” Please explain how you categorised participants who underwent sufficient therapy and experience full remission of ADHD symptoms at the time of the survey.
  • Do all assessments apply for both samples in the same way? Most importantly, do all measures that yield in the categorization to clinical groups, were administered the same way for both samples?
  • For all measures, please provide data on psychometrics and state how they were administered (e.g., via self-report or structured interview, in person or via telephone, with the child or the parent).
  • Since your socioeconomic status seems to be derived from the parents’ education level and profession only, consider calling the variable “parents’ education level and profession”. Otherwise, please state briefly how you calculated your SES score.
  • Please be more specific when describing your statistical analyses. Consider numbering your study aims (referring to the last paragraph of the Introduction) and stick to this structure throughout the whole manuscript. Further, I would expect the description of the PCA in this paragraph. Did you control for multiple testing within the ANOVA/ANCOVA and the Chi square analyses? Please state the applied significance levels.
  • “All children received a score for the three dietary patterns measured.” (150,151): Please elaborate how you calculated this “dietary pattern” score.

Results:

  • “Most of the sample was autochthon and had a similar socioeconomic status” (181,182): Please be more specific when describing results. Briefly explain the context of “autochthon” for your population or consider a more commonly-used wording.
  • “in relation to the control group” (185): Please state the exact control condition you are referring to.
  • Table 1 appears difficult to read and to understand, please reformat.
  • The merit of comparing the food consumption of the present study’s sample with the SENC recommendations seems to decrease even further, because you not only unified your samples but also your age groups (206). The figures simply show that children and adolescents in general do not adhere to dietary recommendations in most food groups. The age-specific role of ADHD remains either unclear or irrelevant.
  • “In the oldest age group” (224): Please be more specific.
  • When describing significant differences in dietary pattern scores within clinical groups (244-248), please avoid reporting differing directions of associations and consistently report, for example, which clinical group had higher scores in a specific dietary pattern compared to which other clinical group (rather than reporting lower scores occasionally).

Discussion:

  • Please revise the sentence: “However, any of them did not show remarkable differences in the food intake between those with ADHD and the controls.” (267,268)
  • In the following paragraph (270-283), you address the important issue of bias due to ongoing therapy, specifically due to pharmacological treatment. Consequently, you controlled for pharmacological treatment (173), which had no impact on the results on food consumption (203,204). Did you also apply control variables on the analyses of dietary pattern scores by clinical group (which are shown in Table 3)? If not, please explain. If controlling for ongoing pharmacological treatment yielded in the same results, the explanation of differing results to Rios-Hernandez et al. due to different levels of medication in the two studies, becomes less compelling. Further, at a later point, you state: “In contrast, Bowling et al [26], did find that their participants continued eating in an unhealthy way despite their medication.” (328,329), which again blurs the understanding of the impact of (treated or untreated) ADHD on eating patterns. Consider explaining this aspect in far more detail.
  • Please thoroughly check the text for misspellings or other orthographical errors (such as in “[…] Wester-like patterns Henriksson et al […]”, 304).
  • Your interesting result that the inattentive subtype of ADHD might have the strongest association to unhealthy eating patterns might be described more prominently and explained more clearly (308-315) – even more as it is contradicting your hypothesis.
  • Drop the term “risk of ADHD” (319) as it implies causal relationships. You provide one other prospective study (Mian et al, 339), in which authors concluded that unhealthy eating behaviour did not increase “a risk of ADHD appearing”. The design of your study, in any case, does not allow to draw conclusions on the prospective or potential causal relationship between unhealthy eating and ADHD. Thus, I would recommend to omit all sections regarding this aspect.
  • As mentioned above, while it is an interesting fact (and highly relevant for public health issues) that the observed children and adolescents do not adhere to dietary recommendations, it does not contribute to your main research aims.
  • Limitations: You correctly state your major limitations. Since your data on food consumption seems to be not only self-reported, but also self-reported by an external observer (the parent), there might be a reporting bias. Further, self-reported data on food intake is widely known for its biased nature. This might have effects on the interpretation of your results and should be accounted for in the limitations section.
  • “Future research with prospective studies could give us more information about whether the dietary patterns followed by children with ADHD are the cause or the consequence of it.” (378-380): As mentioned above, both your study design and the literature presented do not seem suited to draw conclusions or even suggestions regarding “causes or consequences” (i.e., the prospective or causal relationships between unhealthy eating behaviour and the risk of ADHD). Thus, I recommend to remove all sections on this specific aspect.

Reviewer 2 Report

This study aimed to determine the food consumption and dietary patterns of preschool children and school-age children with and without ADHD. The authors found that the ADHD group had less adherence to the healthy pattern than the control group, and it was positively associated with the Western dietary pattern. Although the paper is well written, some main drawbacks in this study restrict the scientific merits.

  1. This study has significant inherent limitation due to its nature of the case-control study design and bottle-neck in dietary data assessment. This study could not conclude the causal relationship between poor dietary pattern, nutritional status, and ADHD.
  2. This study lacked of data about nutritional biochemistry profiles, and the dietary patterns were only assessed using questionnaires.

Round 2

Reviewer 1 Report

The authors have been highly responsive to my comments and I found the manuscript to be greatly improved.

Reviewer 2 Report

The manuscript has been substantially improved.